# Eco-Friendly Clothing Market: A Study of Willingness to Purchase Organic Cotton Clothing in Bangladesh

**Md Mehedi Hasan** [1] , **Liling Cai** [2,*] , **Xiaofen Ji** [2] **and Francisca Margarita Ocran** [3]

1 School of Fashion Design and Engineering, Zhejiang Sci-Tech University, Hangzhou 310058, China; mehedi.mehedi00@gmail.com
2 Department of Fashion Design and Engineering, School of International Education, Zhejiang Sci-Tech University, Hangzhou 310058, China; xiaofenji@zstu.edu.cn
3 School of Textile Science and Engineering, Zhejiang Sci-Tech University, Hangzhou 310058, China; margariocran@gmail.com
\* Correspondence: caililing@zstu.edu.cn

**Abstract:** This research study aims to achieve a developing country's sustainable development in the clothing industry by exploring consumer behavior to a willingness to purchase organic cotton clothing (OCC) and apparel retailers' responsibilities. Organic cotton clothing development in the fashion industry could play an important role in textile and environmental pollution and create new business opportunities for green clothes. Survey data was collected from top cities in Bangladesh, and 303 useable responses were collected (81.5% male and 18.5% female). In our survey, 60.7% of the participant was employed. This research model was inspired by the Theory of Reasoned Action (TRA) and added some new variable that influences purchase willingness under OCC fashion in developing countries like Bangladesh. The findings of this study stated that consumer environmental concerns and consumer attitudes positively impact the OCC purchase willingness of Bangladeshi consumers. Also, the authenticity and fashionable of OCC products have a significant impact on Bangladeshi consumer purchase intention. Product Performance found an indirect effect on Bangladeshi consumers' intention. Furthermore, this study will find that the Bangladeshi market is already very aware of the sustainability movement and concerned about environmental issues. Retailers should focus more on environmental awareness of OCC textile and authentic OCC items. Also, this study will update previous research findings on consumer attitudes toward OCC fashion in the Bangladesh market.

**Keywords:** organic cotton clothing (OCC); purchase willingness; brand responsibility; sustainable clothing

## 1. Introductions

Different types of unsustainable consumption patterns in today's world face several environmental problems such as pollution, greenhouse gas, global warming, etc. It's now become a global issue [1]. The textile and fashion industry is one of the environmental polluters in the industrial sectors [2]. For covering our bodies from weather and other social purposes, human beings use natural materials found in the environment. Nowadays, these natural materials are used in different products and have different types of applications. Cotton is the most widely recognized fiber as well as widely used raw material in this textiles industry. Except for a few artificial fibers, this industry is based on different types of cotton. We get cotton through our agriculture system. In the modern days, many things are involved in our agriculture system that has serious effects on the environment. Our cotton consumption increases every year, with world cotton consumption growing 2.66 percent in the 2021-22 seasons [3]. Naturally grown cotton does not harm the environment unless fertilizers, pesticides, and other harmful chemicals are used [4]. But now, the use of fast-growing cotton and the production of traditional cotton fiber using various chemicals

harms the soil, water, and air, resulting in pollution of the environment [5]. Organic agricultural practices advance a more secure, eco-friendly and do not harm living creatures; organic fertilizers, natural pesticides, and insecticides are used [6]. So, organic cotton is much better for farmers, consumers, and all living things. According to the US Census Bureau, the current world population is 7.8 billion [7], and UN DESA predicted 8.5 billion by 2030 and 9.7 billion by 2050 [8]. According to current solutions, people need to have cloth and cotton consumption, and demand will increase. One of the most challenging issues is global warming and climate changes directly connected with CO2 emissions [9]. 1.22 to 2.93 billion tonnes of $CO_2$ are added to our environment by textile industries. The carbon footprint of cotton is a remarkable top, around 2–4 t per hectare. On the other side, compared to traditional cotton, organic cotton has 40% less "global warming potential" and suggests a 91% reduction in natural water consumption [10]. In this situation, we strongly believe Organic cotton clothing practice can be one of the great solutions for this industry, making it more sustainable and environmentally friendly. Not only in western developed countries but also in developing countries like Bangladesh. Because many recent studies on sustainable development and the green movement are based on western developed countries while less intention on developing countries such as Bangladesh [5]. Sustainable fashion practice offers a way to solve many environmental problems related to production and fashion consumption [11].Recent studies on consumer willingness to purchase behavior find that growing awareness and environmental concerns influenced consumer purchase decisions on organic clothing cotton [12]. We believe that consumers can influence the transformation of fashion companies towards sustainability through their purchasing decisions [13]. Also, the fashion industry should be aware of environmental safety, human safety, and other remarkable corporate social responsibilities. A study on European consumers believed that fashion brands should take the challenge this climate change and environmental protection [5].In many developed countries, fashion retailers focus on organic clothing products to increase their market share [14]. Many renowned brands and retailers were making a profit from organic clothing. Brands like Nike, H&M, C&A, and Wal-Mart have introduced 100% organic materials [15]. '2025 Sustainable Cotton Challenge' from May 2017 textile industry motivate clothing brands and retailers to make 100% of the cotton they use come from sustainable sources by 2025 [16]. Clothing Companies should reduce their environmental carbon footprint; in many ways, ethical sales are growing but not enough categories in the market [12]. Consumer willingness depends on many things; relevant literature from previous research summarizes that high product costs, less choice, aesthetic difficulties, the credibility of information, and uncertainty about the actual environmental benefits were the main limitation for consumers to purchase eco-friendly products, including clothing [12]. Consumer demand for green products is also connected with price and income strategies, especially green products on the market sold at a higher price than regular products (Wilson and Ilartsen, 2010) [17]. However, nowadays, environmentally concerned consumers are growing and they are shown to pay a premium price for eco-friendly products [18]. Boks And Stevels indicate that consumers are willing to buy green products when their income and budget increases [19]. A study on organically grown products (OGPs) already proved that monthly household income is statistically significant and positively influences the consumer purchase intentions of organically grown food products [20].

There are a lot of studies available on consumer behavior toward Organic personal care products and organic food products. But only a few studies are available on OCC purchasing behavior intention [14]. Therefore this study inspired by The Theory of Reasoned Action (TRA), also modified and extended other essential variables that impact willingness to purchase intention OCC. TRA is an old model, but many researchers of consumer behavior-related studies inspired their research fields and questionnaires on this model. TRA model is widely used and extended in many ways [21,22].

*Bangladesh Fashion Market*

Economic growth changes rapidly, and most manufacturing units are developed in underdeveloped countries. Environmental issues are not a priority in developing countries, but they are a significant threat to the world [4]. World population ranking Bangladesh is now world rank 8 [23], besides Bangladesh also 2nd largest clothing manufacture whole over the world [5]. According to the world bank, this COVID-19 limitation Bangladesh's economic growth rates pick up to 6.9 percent in the year 2021, and last decade fastest-growing economy in the world [24]. According to overall economic growth, Bangladesh's domestic market size is also increasing, and some global brands are already active in the Bangladeshi market. This includes the German leading sports brand Puma, and the popular Japanese brand Uniqlo which started their retail business in Bangladesh a few years ago [25]. Similarly, in the last decades, several local brands have had strong positions in the Bangladeshi market like 'sailor', 'yellow', 'Aarong' and other brands which are doing well, and these brands are also undertaking sustainable initiatives. For example, 'sailor' and 'Aarong' are introducing natural cotton examples: organic cotton, Bamboo fiber and Jute fiber products. Therefore, organic products and their impact on the environment has always been an important research topic, although less focus is placed on studying consumer purchase behavior in developing countries.

## 2. Literature Review and Hypothesis Development

Today's generations are more concerned word is sustainable fashion, sustainable fabrics, and natural fibers. Abrar, M investigated consumer purchase behavior of green organic textile products in developing countries such as Pakistan. This study finds that consumers have a positive attitude toward purchasing organic products. The study suggests that retailers create consumer attitudes towards organic clothing products and increase consumers' purchase intention for organic clothing products [14]. Hae Jin Gam also studies mothers' willingness to purchase OCC for childrens' product segment. The result found that mothers' environmental concerns significantly impact their involvement in OCC, significantly influencing mothers' willingness to purchase OCC [12]. In a case study on organic cotton clothing in Hawaiian consumers, Lin's also discussed and found that consumers are willing to pay more when consumers are more concerned about protecting the environment or involved in environmental issues [26]. Gwendolyn Hustvedt [27], has shown that consumers are motivated by their belief in the benefits of buying organic clothing. It is clear that it explains how organic cotton clothing supports organic farming; it is another way to promote the environmental benefits of buying organic cotton clothing. Previous studies on similar topics, green eco-friendly marketing, and consumer purchase behavior mentioned that intricacy of information, product advantages, functional benefits and positioning of green products, and the celerity of environmental effects benefits were the crucial barriers to purchasing eco-friendly products [28].

*2.1. Theory of Reasoned Action (TRA)*

The theory of reasoned action (TRA) used to predict and understand consumer behavior, and their behavioral intention developed by Ajzen and Fishbein 1980 [29] is a belief-attitude-behavioral intention model. In the TRA model, when consumers implement a purchasing activity, they benefit from that behavior and gain approval from others. TRA seeks to predict consumer purchasing and intent [21]. TRA theory has been extensively accepted and strongly used in multiple studies such as marketing, consumer motivations, promoting recycling behavior, the intent to engage in sustainable behaviors, and the attitude toward luxury fashion goods [29]. Researchers also modified and added the TRA model to their research plan. A recent study's effects of government subsidies on green smartphones successfully used the TRA model and added new variables [21]. We will also find some variables that also affect consumer buying behavior. This study is also an expansion of TRA. In addition, consumer environmental concerns, fashionable products in organic segments, product authenticity, performance, and consumer economic conditions are also important

research values in this study. Environmental concern is assessed as a subjective norm in the TRA in this research. This study also introduces brand responsibilities like original and authentic organic cotton products and focuses on fashionable organic apparel that can strongly affect consumer purchasing behavior.

### 2.2. Environmental Awareness and Concerns

Environmental awareness and concerns have obtained great attention from consumers all over the world. When consumers are conscious of environmental issues, they raise concerns about the environment and human life. Previous researchers have also found that consumers more aware of the environment are more likely to be involved in ethical purchases. Another research paper also investigated environmental concerns related to ethical behaviors and consumers' willingness to pay more for green electricity [30]. In a case study on organic cotton clothing in Hawaii, resident consumers found that when people are aware of protecting the environment, they have more intention to purchase OCC items [26]. In this study, we similarly investigate in our target consumers whether environmental concerns have affected consumers' purchase intentions under OCC buying intentions.

**Hypothesis H1.** *Environmental concerns have a significant positive impact on willingness to purchase OCC.*

According to the TRA model, attitudes affect the intent to engage in behavior, which affects actual behavior. Attitude is the level at which a person has a favorable or unfavorable opinion of behavior [30]. Many studies have already proved it in different research fields. Kim and Chung found that attitude influenced consumer purchase behavior for organic skincare items [31]. In previous studies, Yan, Hyllegard, and Blaesi also mentioned that attitude predicts consumer purchasing intent for ethically produced fashion items [30]. Therefore, we assume that the results may be similar. Thus, we suggest the below:

**Hypothesis H2.** *Attitudes have a significant positive impact on willingness to purchase OCC.*

### 2.3. Brand Responsibilities in Eco-Friendly Fashion

Brand responsibilities and awareness allow the retailer to provide a long-term ethical plan. It is also an important marketing strategy to attract consumers, especially in the eco-friendly market. Eco-friendly brands also benefit if they are promoted with an image of nature with environmental and social responsibility [32]. In previous research works, it has been found that ethical fashion and style are related important factors for making a decision, and was also mentioned by consumers willing to boycott unethical brand leaders [33]. Some consumers have a different idea about OCC, and they still doubt its quality, finishing, and design. It is common for consumers to expect something good from green products when they make purchasing decisions [32]. This study investigates whether the fashion and trend of an organic apparel product affect consumers' purchase intention. Therefore, in this research, we proposed that:

**Hypothesis H3.** *Fashionable and trendy products have a significant positive impact on willingness to purchase OCC.*

Consumers who want to buy organic cotton clothing are also expected to focus on the performance of the clothing. The previous research found that consumers who used "organic" as a criterion when purchasing clothing products believed that a quality product is a result of purchasing their organic cotton clothing [34]. Product performance and quality are important variables that influence consumers' decisions to purchase. When the consumer is not satisfied with the product's effectiveness, it evaluates the purchasing experience as a time risk [35]. Thus, in this research, we proposed that:

**Hypothesis H4.** *Product performance has a significant positive impact on willingness to purchase OCC.*

Some clothing brands claim their product are organic, but most of the time, they do not carry any certified organic label. Previous researchers have already noted that certified environmental claims can enhance consumers' confidence in the environmental performance of a product and the quality of the organic label product [34]. In addition, some consumers have doubted the authenticity of the product. This study investigates whether products authenticity affects consumers' purchase intention of eco-friendly OCC products. We proposed that:

**Hypothesis H5.** *Product authenticity significantly impacts willingness to purchase eco-friendly OCC fashion.*

*2.4. Purchasing Power of "Green Clothes"*

Consumer purchasing power and economic growth are also important factors in their purchase decision, especially organic clothing, as many researchers mentioned that green products are usually available in the market at a higher price, and when consumers' income increases, their shopping budget also increases. Researchers point out that consumers are more interested in purchasing green products when their income and budget increase [32]. Consumers are willing to pay more for green and renewable products when they are aware of the environmental advantages [30]. Therefore, in this research, we proposed that:

**Hypothesis H6.** *Consumer economic growth and condition can significantly influence consumers' willingness to purchase eco-friendly OCC Fashion and willingness to pay premium purchasing.*

Based on the above literature review and theoretical analysis, this study proposes the following conceptual research model. The model is illustrated in Figure 1.

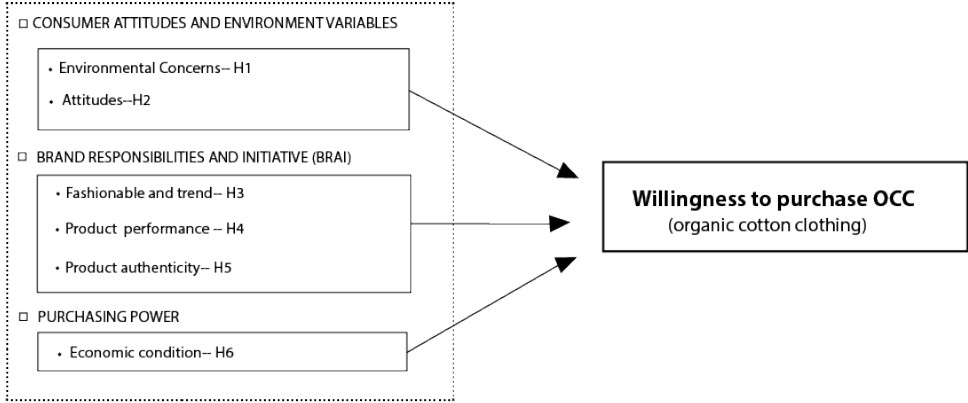

**Figure 1.** Conceptual model of this study.

**3. Methods**

The choice of research methods must be fair and appropriate to the research problem and should be selected based on the study objectives. The online survey was taken with more than 303 adult consumers. We believed that a quantitative research method using a survey is the best way to collect basic information from Bangladeshi consumers. Final data collection and all data evaluation will be IMB SPSS Statistics.,version 22, Release 22.0.0.0,64-bit edition (Hangzhou, China). Our survey data was examined through a reliability and validity test, factor loading, Cronbach's Alpha, and other reliability tests were done. The comprehension of participants' questions from a survey may differ from what the researchers anticipated. It is perhaps impossible to answer the research question evaluation. At that point, the validity of the construction and its reliability needs to be verified [36,37]. Understanding both directions, as well as the strength of the relationship between the research constructs within this study, a Pearson correlation analysis was also done. Pearson's method of correlation evaluates the linear relationship between two variables. The coefficient can be between $-1.00$ to $1.00$, where $-1.00$ shows a negative

relationship, 0 shows no relationship, and 1.00 is a positive relationship. After confirming that the coefficient of a reciprocal relationship is statistically significant, the strength of the relationship can be discussed and measured [37,38]. Finally, a regression analysis is done for understanding the impact of all variables because researchers utilize regression coefficients and their significance to estimate the importance of each construct. The most commonly used is multiple simultaneous regressions, where all IV regression calculations are taken at once. The regression coefficient gives an analysis of the direct impact of each IV on the DV when considering other IVs [39].

### 3.1. Question Design

This research questionnaire was adopted from previous research and a few of the author's questions according to the research direction [5,15,32,37,40–42]. Based on an online survey written in English. The first few questions are consumer demographic questions that determine their social position. The other question is to determine their involvement, knowledge, daily activities, attitude toward OCC, and willingness to purchase on OCC. Multiple-choice questions and a rating scale plan the question layout for this survey. This survey asks for the responses to each item question is placed on a 5-point Likert scale with "1 = strongly disagree, to 5 = strongly agree" [5].

### 3.2. Sample

This study was conducted quantitatively using the online survey method, Google online survey form used, and distributed survey links mainly using social networks. The survey was distributed to a random sample of more than 303 young consumers. Overall, 303 valid questionnaires have been completed. All questionnaire surveys were self-administered to respondents who lived in the top two cities in Bangladesh. Out of 303 responses, 81.5% were male consumers, and 18.5% were female. Almost 60.7% of responders were employed, and the rest of the, worked in business, were unemployed, and were students. The demographic information of the participants is presented in Table 1.

**Table 1.** Demographic information of the sample.

| Variable | Description | Percentage (%) |
|---|---|---|
| Gender | Male | 81.5 |
| | Female | 18.5 |
| | Others | 0 |
| Age | <18 | 1.3 |
| | 19–25 | 28.1 |
| | 26–35 | 64.4 |
| | 36–45 | 5.9 |
| | >46 | 0.3 |
| Education | Below undergraduate | 5.3 |
| | Undergraduate | 28.4 |
| | Graduate | 46.5 |
| | postgraduate | 19.8 |
| Employment level | student | 20.8 |
| | Employed | 60.7 |
| | Unemployed | 6.3 |
| | Business holder | 11.2 |
| Income level | <14,000 BDT | 21.1 |
| | 15,000–25,000 BDT | 19.8 |
| | 25,000–50,000 BDT | 30 |
| | 50,000–100,000 BDT | 22.1 |
| | >100,000 BDT | 6.9 |

## 4. Results Analysis

Many scholars mentioned that a coefficient of 0.7 or above represents the threshold for reliability. Others believe that values of 0.6 or above are also acceptable [37]. Our Cronbach's Alpha is 0.851 and KMO is 0.830; both are reliable and suitable for our studies, presented in Table 2.

**Table 2.** Cronbach's Alpha Reliability Statistics.

| Reliability Statistics | | |
|---|---|---|
| **Cronbach's Alpha** | **Cronbach's Alpha Based on Standardized Items** | **N of Items** |
| 0.851 | 0.853 | 33 |

Table 3 shows the results of this analysis. Most of the correlations were significant at the *p* level of <0.01. However, product performance (PP) correlates with purchase intention (PI) at the significance level of $p < 0.05$. There are some differences also; some correlations did not have statistical significance, such as FT and PI. Great correlation strength is between FT and PP with an r of 0.635. In the other EC, AT, PA, and Econ constructs, this variable significantly correlated with all other constructs.

**Table 3.** Inter-correlations for all constructs.

| Variables | | EC | AT | FT | PP | PA | ECon | PI |
|---|---|---|---|---|---|---|---|---|
| EC | Environmental concern | —— | 0.404 ** | 0.314 ** | 0.244 ** | 0.404 ** | 0.364 ** | 0.361 ** |
| AT | Attitude | — | —- | 0.292 ** | 0.258 ** | 0.448 ** | 0.509 ** | 0.483 ** |
| FT | Fashionable and trend | — | — | — | 0.635 ** | 0.379 ** | 0.295 ** | 0.111 |
| PP | Product performance | — | — | — | — | 0.321 ** | 0.271 ** | 0.115 * |
| PA | Product authenticity | — | — | — | — | — | 0.578 ** | 0.526 ** |
| ECon | Economic condition | — | — | — | — | — | — | 0.592 ** |
| PI | Purchase intention | — | — | — | — | — | — | — |

** Correlation is significant at the 0.01 level (2-tailed). * Correlation is significant at the 0.05 level (2-tailed).

The independent variables significantly predict consumer purchase intention (PI), $F_{(6,296)} = 42.336$, $p < 0.001$, which indicates that our independent factors are an impact on consumer purchase intentions. Moreover, The $R^2 = 0.462$ depicts this model explaining 46.2% variance in purchase intention. Table 4 shows the summary of the findings.

**Table 4.** Model Summary.

| Model | R | R Square | Adjusted R Square | Sig. F Change |
|---|---|---|---|---|
| 1 | 0.680 [a] | 0.462 | 0.451 | 0.000 |

[a] Predictors: (Constant), EConv, PPv, ECv, ATv, PAv, FTv.

Table 5 reveals that applying all the IVs at once creates a multiple correlation coefficient ($R^2 = 0.462$). It implies that 46.2% present of the variance in purchase intention of organic-cotton clothing can be explained by all constructs taken together. The standardized beta coefficient (ß) is considered to compare each IV's influence size directly. Table 5 shows that economic condition (Econ, ß = 0.362) has the greatest impact on PI, product authenticity (PA) (ß = 0.259), AT (ß = 0.198), FT (ß = 0.158) and EC (ß = 0.105). Econ's positive sign of ß means that a 1 standard deviation (SD) increase in Econ's generates a 0.362 SD increase in the predicted PI. The *p*-value facing every IV points out if that variable is important, adding to the calculation for explaining PI from the entire group of IVs. As a result, Econ (*p* = 0.000), PA (*p* = 0.000), AT (*p* = 0.000), FT (*p* = 0.006) and EC (*p* = 0.034) are the sole constructs that are importantly contributing to the explanation.

**Table 5.** Regression analysis overview for all constructs.

| Hypotheses | Regression Weights | ß | T | *p*-Value | Hypotheses Supported |
|---|---|---|---|---|---|
| H1 | EC > PI | 0.105 | 2.130 | 0.034 | Accepted |
| H2 | AT > PI | 0.198 | 3.779 | 0.000 | Accepted |
| H3 | FT > PI | −0.158 | −2.747 | 0.006 | Accepted |
| H4 | PP > PI | −0.043 | −0.765 | 0.445 | Rejected |
| H5 | PA > PI | 0.259 | 4.626 | 0.000 | Accepted |
| H6 | ECon > PI | 0.362 | 6.463 | 0.000 | Accepted |

$R^2 = 0.462$, F (6296) = 42.336, $p < 0.001$.

We provide a summary of this study's findings, both accepted and rejected hypotheses. This study examines the factors influencing Bangladeshi people's purchasing intention toward eco-friendly organic cotton clothing (OCC). In addition, the authors strive to find the effects of those factors, including environmental concern (EC), attitude (AT), fashionable and trend (FT), product authenticity (PA), and economic condition (Econ), as well as product performance (PP). The overall model generated from linear regression analysis is statistically significant. The findings of our studies are consistent with many previous studies.

In this study, H1 EC toward eco-friendly OCC was found to have a positive effect on PI, with a standardized regression coefficient (ß) of 0.105 ($p = 0.034$). Past research examined influencers of PI toward eco-friendly OCC among American mothers' willingness to purchase OCC [12]. It was revealed that the EC is significantly influenced by consumer involvement in OCC, and the effect was found to be positive (ß) of 0.33 ($p = 0.04$). In addition, another researcher examined factors motiving male consumers Awareness of the environmental consequences has a positive impact on male consumers' engagement in eco-friendly clothing. Its results are also positive and (ß) of 0.180 ($p = 0.001$) [2]. In addition, a study on purchase intention towards OCC and the result shows that EC had a insignificant positive impact on purchase intention (β) of 0.216 ($p =0.168$) [14]. The results of Table 5 show that there was a significant relationship between the environmental concern variable and the purchase intention in OCC ($p = 0.034$, $R^2 = 0.462$). The *p*-values were positive and statistically significant, indicating that participants with greater environmental concerns showed a higher intent to purchase OCC. Therefore, H1 was accepted. Furthermore, H2 is confirmed as AT significantly and positively influences the PI, with a standardized regression coefficient (ß) of 0.198 ($p = 0.000$). Previous research studies on purchasing green apparel products found that attitudes will positively influence the consumer purchasing intent towards green clothing products and it was found to be significant and β = 0.17 for American consumers similar to beta β value with our studies [43]. In addition, another examined factor motiving on male consumer and AT had a significantly positive impact on male consumers for engaging in eco-friendly clothing. Its results were also positive and (ß) of 0.284 ($p = 0.000$) [2]. A past researcher also examined influencers of PI toward eco-friendly clothing among American university students. It was revealed that the PI is significantly determined by AT, and the effect was found to be positive. Similarly, in purchasing bio-cotton apparel among female Korean households items, researchers found AT as a significant and positive influencer (ß = 0.34) [37]. There are some differences, however; recent surveys of the USA and Bangladeshi consumers' organic cotton clothing purchase behavior comparative studies found that AT is not significant in PI toward OCC purchases in the Bangladesh market. With a standardized regression coefficient for Bangladesh consumer (ß) of 0.19 ($p = 0.349$) [5].We can argue this point because this result variation may be due to the number of participants. The studies were based on only 51 participants from the Bangladeshi market. Moreover, it has been shown that fashionable and trendy products (FT) have a significant impact on PI (ß = −0.158) ($p = 0.006$), which supports H3. In addition, the negative sign indicates that more perception of limited availability regarding eco-friendly bio cotton clothing is associated with lower degrees of PI. A previous

study already investigated consumer fashion consciousness having a positive impact on purchase intention of fast fashion products with (ß = 0.375) (*p* = 0.000) [44]. Previous researchers also found that fashion and style are related factors in making decisions to purchase clothes [33]. Fashionable products are among the trendiest items, and consumers share and find the latest trend in real-time [45]. In our studies, we also find that fashionable products have a significant impact on PI. Therefore, a lot of large retailers may have to rethink their design practices according to eco consumers' requirements and introduce more organic cotton clothing.

Next, product authenticity (PA) was found to have a significant influence on PI (ß = 0.259, *p* = 0.000). H5 is supported. In addition, the positive sign indicates that more positive PA is related to greater degrees of PI. Previous studies also mentioned that the effect of authenticity on purchase intention is significant [46]. Recent studies also found that product authenticity had a positive impact on PI (β = 0.221) for Iraqi Facebook users. It is recommended that marketers should provide practical and honest information in their promotion messages [47].

In this study, economic condition (Econ) toward purchasing organic cotton clothing was found to be the highest of all constructs and has a positive effect on PI, with a standardized regression coefficient (ß) of 0.362 (*p* = 0.000). The results in Table 5 show that consumer purchasing power and consumer economic growth positively influence consumer purchase intention. Therefore H6 was accepted. Consumer economic growth and income always influence purchase intentions. Previous studies have shown that high-income respondents have a strong desire to purchase through social media websites [48]. However, a survey of Pakistani consumers found that there was no relationship between income and purchase intentions [49].

Though another hypothesis in our model, product performance (PP) had a relation in our correlation model, H4 was not accepted in consumers' PI toward green OCC products in the Bangladesh market. Product performance (PP) with a standardized regression coefficient (ß) of −0.043 (*p* = 0.445). In our studies, both the lowest Beta (ß) values and *p*-value are more than our acceptable level. When we surveyed organic products that are not durable, 29.4% of participants replied neutral and 21.1% disagreed. The same participants were asked another survey question about organic cotton styles and fashion, 32.3% agreed, and almost 15% strongly agreed that organic cotton did not fully fit their style and fashion. Therefore, in that survey, we can point out that consumers in the Bangladesh market are more concerned about fashion and style than product lifetime. Finally, we determine the effect of environmental concerns, attitudes, fashionable products on organic clothing segments, product performance, and economic growth on the purchase intention of consumers of OCC products. This study will find that the Bangladeshi market is already very aware of the sustainability movement and concerns about environmental issues. Bangladeshi consumers show a positive attitude toward OCC items. Bangladeshi markets have a good potential for all kinds of organic cotton clothing. Retailers should be concerned and take the initiative in their eco-friendly clothing design practice, also take the initiative and emphasis the values of organic cotton in our environment and human life. Fashion retailers should be concerned promote honest and transparent information to consumers because product authentication is one of the important factors in consumer purchase decisions, especially when consumers are aware and willing to purchase eco-friendly OCC items. This research can be conducted in different sectors and countries using the qualitative method through focus groups. This study will be beneficial for the environmental improvement and economic growth efforts among policymakers, our academy, and all over the textile industry. This study enables them to design strategies that ensure interaction with consumers to promote desirable behavior towards organic cotton clothing products. Figure 2 illustrates the effect of IV on dependent variables (e.g., PI). Arrows with a solid line represent a statistically significant effect when accompanied by them; the dashed line shows a non-significant effect.

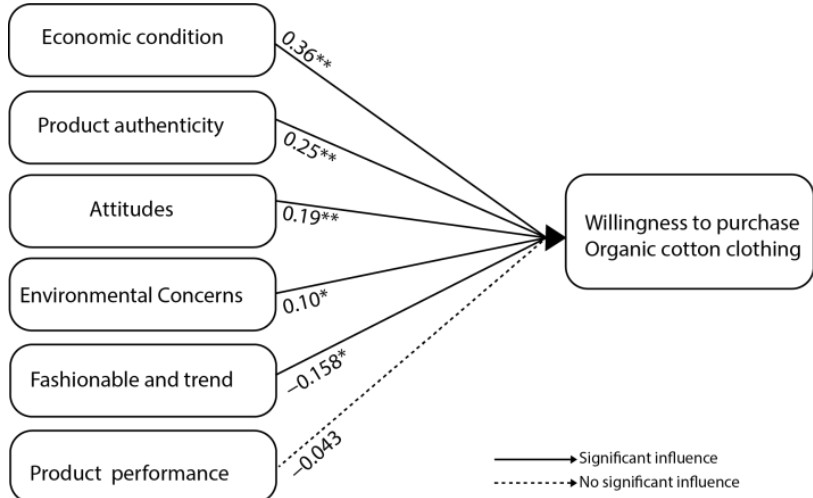

**Figure 2.** Summary of revised model. All values are standardised regression coefficients. ** $p < 0.00$, * $p < 0.05$.

## 5. Limitations and Future Research

This study was conducted in the top two busiest metropolises in Bangladesh, Dhaka, and Gazipur. Bangladesh is a country of population density. Future research can examine both urban and rural contexts such as income by accurately presenting different population profiles such as age, level of education, and population. Furthermore, this research work can be replicated in other developing countries, and future research should include other theories or methods. Future researchers could verify the proposed research model on other products such as home textiles and luxurious items that are made from organic cotton or other sustainable textiles.

Research survey items on "Willingness to Purchase Organic Cotton Clothing in Bangladesh" added below in Appendix A Table A1.

**Author Contributions:** Conceptualization, M.M.H. and L.C.; methodology, M.M.H. and L.C.; software (SPSS), M.M.H. and F.M.O.; formal analysis, M.M.H.; resources, M.M.H.; data curation, M.M.H. and F.M.O.; writing—original draft preparation, M.M.H.; supervision, L.C.; project administration, X.J.; funding acquisition, X.J. All authors have read and agreed to the published version of the manuscript.

**Funding:** The paper is supported by Zhejiang Provincial Philosophy and Social Science Planning Project (21NDJC062YB), National Social Science Foundation of China art program (20BG134).

**Institutional Review Board Statement:** Not applicable.

**Informed Consent Statement:** Informed consent was obtained from all subjects involved in the study.

**Data Availability Statement:** The information presented in this study is available in the manuscript. Additional data is available at the request of the author concerned.

**Conflicts of Interest:** The author declares no conflict of interest.

## Appendix A

**Table A1.** Research Survey item and sources.

| Variable | Item | Source |
|---|---|---|
| Environmental concern | It is time for environmental groups to get more radical.<br>I am extremely worried about the state of the environment.<br>We are in a serious negative impact of environmental issues.<br>I feel personally helpless to have much impact on the environment. | [40,42] |
| Attitudes | Buying organic cotton clothing instead of conventional clothing would feel like the morally right thing to do.<br>Buying organic cotton clothing instead of conventional apparel would make me feel like a better person.<br>Buying organic cotton clothing instead of traditional clothing will feel more beneficial to the environment. | [15] |
| Fashionable and trend | In my opinion, organic cotton clothing is only popular at the moment.<br>Organic cotton clothing does not come in different styles that fulfill my need.<br>In my opinion, organic cotton clothing is not trendy and fashionable. | [37,41] |
| Product Performance | organic cotton clothing is of good quality and has a better lifetime.<br>Organic cotton clothing is not as high-quality then conventional cotton clothing.<br>Organic cotton clothing will not be durable when cleaned (e.g., color fades, form changes) | [37] |
| Product authenticity | I suspect that I can't distinguish real organic cotton clothing.<br>I only believe in organic cotton garments that have been authoritatively certified.<br>I will be more likely to purchase eco-friendly OCC fashion if their apparel Brand makes sure their product is Authentic.<br>Not all businesses claiming to sell organic cotton clothing are credible<br>Those businesses that often advertise environmental friendliness do not all sell real organic cotton clothing<br>I sometimes don't believe that some businesses claim that their clothes are organic cotton clothing. | [15,42] |
| Economic condition | The cost of organic apparel was the same as nonorganic apparel. I would be more likely to buy more organic apparel.<br>I am willing to pay more for eco-friendly products (i.e., green, organic).<br>Is the current financial development and growth affecting your shopping behavior?<br>According to financial development, did you concern that clothing and shopping expenses are increased? | [15,32,41] |
| Purchase intention | I would gladly buy more organic cotton apparel if I could find it easily.<br>I desire to buy organic cotton clothing if it's health and environment friendly.<br>I am willing to pay more for organic cotton clothing if it's fashionable and fulfills my needs.<br>The next time you go shopping, I will likely purchase organic apparel products. | [5,15] |

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
