# Peer review of "Eco-Friendly Clothing Market: A Study of Willingness to Purchase Organic Cotton Clothing in Bangladesh"

_sustainability, doi:10.3390/su14084827_

Round 1
Reviewer 1 Report
English must be improved, is poor. Several times is mentioned that research results belong to thesis, what isn´t fine.
Methodology is explained, but isn´t mentioned scientific literature to support it.
Beginning of the results topic must be moved to the methodology
Author Response
Dear Reviewer,
According to your comments and suggetiones, we have done below correction:-
1. English must be improved, is poor. Several times is mentioned that research results belong to thesis, what isn´t fine.
Authors response: Thanks for your suggestion. We revised it and try our best to correct grammar, choice of words, and the Formality level. Also, add some points and detailing according to the referee's suggestions by articles and papers. We also reduce some previous thesis references.
2. Methodology is explained, but isn´t mentioned scientific literature to support it.
Authors response: we rework and add required scientific literature detailing our methodology topic.
-Now in the methodology part we add scientific literature detailing( 5 references).
-In the question design part we add more references to support it (Now 8 references).
3.Beginning of the results topic must be moved to the methodology.
Authors response: yes, according to your suggestion we moved and add specific result analysis points with reference in our methodology.
Reviewer 2 Report
GENERAL COMMENTS
The language used in this manuscript should be improved
ABSTRACT
Why was the data collected from many cities when the manuscript was about Dhaka?
INTRODUCTION
Poorly written introduction – authors should focus on the right subject. All the numbers need references.
87% imported textiles in Europe imported – this is from Bangladesh, China, India and other countries. Authors are moving out of the topic and do not have focus on their project.
The consumer willingness depends on many things including the cost. I do not see anything of that sort in introduction. This is referred to little extent later in the manuscript.
METHOD
Table 1 should have income as well. This can give a clearer picture
Authors need to improve the manuscript considerably before submitting again.
Author Response
Dear Reviewer,
According to your comments and suggetiones, we have done below correction:-
- The language used in this manuscript should be improved.
Author's response: We revised it and try our best to correct the grammar, choice of words, and Formality level. Also, add some points and detailing according to the referee's suggestions. - ABSTRACT
why was the data collected from many cities when the manuscript was about Dhaka?
Author's response: Data was collected from 2 top cities, Dhaka and Gazipur. Dhaka is the capital city of Bangladesh and Gazipur is the largest city corporation in Bangladesh. Both cities are most populated and we believe both cities are important.
But due to avoid any conflicts we decided to remove a specific city name from our manuscript title. - INTRODUCTION
poorly written introduction – authors should focus on the right subject. All the numbers need references.
Author's response: we revised and re-organized. Also, add some points and detailing according to all referee's suggestions. Also checked and make sure all the specific numbers/data are referenced. - 87% imported textiles in Europe imported – this is from Bangladesh, China, India and other countries. Authors are moving out of the topic and do not have focus on their project.
Authors response: Yes, we rework and removed this type of point, and focused on our topic. Also, add some points and detailing according to all referee's suggestions. - The consumer willingness depends on many things including the cost. I do not see anything of that sort in introduction. This is referred to little extent later in the manuscript.
Author's response: According to this point, we extend our introduction with reference.
METHOD:
6. Table 1 should have income as well. This can give a clearer picture.
Author’s response: yes according to your suggestion we added our income variables to our Demographic table.
Reviewer 3 Report
1. The work described is explained with numerical data for there has been a good study
on the use of organic cotton.
2. There are capitalization mistakes in some parts of the article. These mistakes need to
be reformed. ( page 5; 207 , page 7;295 , page 8;325 , page 10; 393-394)
3. Material method parts consist of only one paragraph. The study could have been
supported by more articles (only 4 articles)
Author Response
1.The work described is explained with numerical data for there has been a good study on the use of organic cotton.
Author's response: Thank you for your evaluations we try to add and explained all available data in our manuscript.
2.There are capitalization mistakes in some parts of the article. These mistakes need to be reformed. (page 5; 207 , page 7;295 , page 8;325 , page 10; 393-394).
Author's response: we worked on your evaluations, you mentioned page by page, it’s really helpful. We corrected all of these issues.
3. Material method parts consist of only one paragraph. The study could have been supported by more articles (only 4 articles).
Author's response: we worked on your evaluations, and we added more information and articles reference’s in our overall methodology and question part.
-Now in the methodology part we add scientific literature detailing( 5 references).
-In the question design part we add more references to support it (Now 8 references).
Round 2
Reviewer 1 Report
Congratulations for authors for imprevement of the article.
Reviewer 2 Report
The language is still poor and it is important that it is improved before publishing online